# An Extended Model of the Theory of Planned Behaviour to Predict Local Wine Consumption Intention and Behaviour

**DOI:** 10.3390/foods10092187

**Published:** 2021-09-15

**Authors:** Edgar J. Sabina del Castillo, Ricardo J. Díaz Armas, Desiderio Gutiérrez Taño

**Affiliations:** Department of Business Management and Economic History, Faculty of Economics, Business and Tourism, University of La Laguna, Camino La Hornera, 37, 38200 San Cristóbal de La Laguna, Spain; alu0100386080@ull.edu.es (E.J.S.d.C.); rjdiaz@ull.edu.es (R.J.D.A.)

**Keywords:** local wine, theory of planned behaviour, consumer ethnocentrism, cosmopolitan

## Abstract

The consumption of local agricultural products boosts the regional economy and employment whilst preserving the rural landscape and environment. In this research, the background of local wine consumption behaviour will be studied, using an extended model of the Theory of Planned Behaviour. Partial least squares structural equation modelling (PLS–SEM) was used to test the hypotheses. The study was conducted in the Canary Islands with a sample of 762 people. The results confirmed a relationship between intention and perceived behavioural control. Furthermore, the ethnocentric personality was found to have a positive influence and the cosmopolitan personality a negative influence. The personal norm and place identity were also confirmed to be related to attitudes towards such behaviour. This study contributes to the literature by adding constructs to this theory that are relevant to local wine consumption. It also addresses the implications for those involved in the marketing of local products.

## 1. Introduction

In recent years, consumer interest in local foods has gradually increased due to consumers perceiving them as products with a wide range of benefits [1,2,3]. One key factor has been the capacity that this has brought about to improve the sustainability of the food system [4], helping to protect the local economy and jobs [5,6], preserve the rural landscape and environment [7], and reduce the use of fuel and the environmental impact associated with transporting food around the globe [8,9,10], among other things.

Within the academic sphere, great attention is being paid to the consumption of local food products from different perspectives. For example, studies by Rainbolt, Onozaka and McFadden [11] and Robinson and Smith [12] successfully applied the Theory of Planned Behaviour (hereinafter TPB) [13] to predict behaviour when choosing local foods. Other studies focusing on this subject have added constructs to the TPB to explain a greater variation in personal conduct [14,15] or have relied on the Social Identity Theory (hereinafter SIT) [16,17] to explain consumer behaviour [18,19].

However, what has not yet been devised is a model that includes both of these theories to predict the consumption of local products, reinforced with influential constructs of the ethnocentric and cosmopolitan consumer personality.

According to Hollebeek, Jaeger, Brodie and Balemi [20], Jaeger, Mielby, Heymann, Jia and Frost [21] and Yang and Paladino [22], where the wine comes from is among the most important selection criteria, which is why this work has focused on wine as a local product. It has been verified that research has been carried out in this area in different parts of the world based on the TPB or variables thereof, in order to study the intention to purchase and consume local wine [23,24,25,26].

In this work, once the factors that influence local wine consumption intention were determined, an extended model was developed based on the TPB and corroborated to check whether these relationships are significant and what impact they have on local wine consumption behaviour.

PLS–SEM was used to corroborate the proposed theoretical model and test the hypotheses. A survey was carried out to gather data from a sample of 762 people obtained via an online panel of adults who were over 18 years of age, frequent wine consumers, and residents of the Canary Islands (Spain).

Analysis of the corresponding data produced a series of relevant results, as well as their implications and limitations, which validate this study and contribute to the literature with a new theoretical model that explains the behaviour of local wine consumption.

## 2. Materials and Methods

### 2.1. Literature Review: Theory of Planned Behaviour

The Theory of Planned Behaviour [13] is an extended model of the Theory of Reasoned Action [27] which proposes a model of how human action is guided. It aims to understand the processes that lead to intentional behaviours.

This theory continues to be one of the most popular socio–psychological models used to understand and predict human behaviour [28] and has been successfully applied to behavioural studies of local product consumption [14,29], wine consumption [30,31] and local wine consumption [25,26].

The TPB postulates that an individual’s intention to act is the closest predictor of their conduct and reflects a person’s willingness to engage in a certain objective behaviour. It is hypothesised that intention is a combination of three belief-based components: attitude, perceived behavioural control and subjective norms, which are analysed in this study.

#### 2.1.1. Attitude towards Local Wine Consumption

Attitudes are a person’s positive or negative opinions about people, events, objects or behaviour, which reflect a series of preferences for the latter and generate positive or negative intentions towards purchasing behaviour [32]. According to the TPB, personal attitude is the main deciding factor in the intention to behave in a certain way and, furthermore, represents the positive or negative individual evaluation of such behaviour [13].

Numerous studies of consumer behaviour in the area of food industry marketing show that attitude towards a product greatly influences consumption intentions [15,25,33,34,35]. Along these lines, some authors [36,37,38,39] have verified that attitudes about local products correlate positively with intention to buy them. That is, the more positive the attitude towards the consumption of local products, the greater the likelihood that people will buy local foods.

Therefore, taking into account previous results and studies on wine consumption habits in specific regions in which most people have a positive attitude towards the consumption of local wine [23,24], the following hypothesis has been constructed:

**Hypothesis** **1** **(H1).***There is a positive relationship between attitude towards local wine consumption and local wine consumption intention*.

#### 2.1.2. Perceived Behavioural Control

Perceived behavioural control is explained as an individual’s perception of their capacity or self-judgement in terms of engaging in a certain behaviour [40]. In other words, it is how the level of complexity is perceived, which is determined by control beliefs in order to reveal an individual’s scope of control over a behaviour [41].

Multiple studies have demonstrated that the greater an individual’s perceived behavioural control, the stronger their intention to engage in the behaviour in question [42,43,44,45]. Shin and Hancer [14] confirmed that, as far as local food products are concerned, PBC has an influence on the intention to buy them. For their part, Capitello, Agnoli and Begalli [46] demonstrated that there is a significant correlation between PBC and wine consumption intention. Furthermore, according to Candan, Aydin and Yamamoto [47], most consumers prefer products from their region, ready availability being one of the most important factors of this preference. Therefore, the following hypotheses are proposed:

**Hypothesis** **2** **(H2).***There is a positive relationship between perceived behavioural control and local wine consumption intention*.

**Hypothesis** **3** **(H3).***There is a positive relationship between perceived behavioural control and local wine Consumption*.

#### 2.1.3. Subjective Norms

Subjective norms are understood to be the perceived social pressure to engage in or refrain from a particular behaviour [40] and represent an individual’s perception or opinion of what others believe they should do [48]. James, Rickard and Rossman [49] defined them as a powerful internal control factor that can easily mould a person’s behavioural intention. Subjective norms are commonly identified as another significant predictor of intention which accentuates the level of importance of the thoughts of another individual [41].

Several specific studies on wine consumption have confirmed that subjective norms have an influence on the behavioural intention of consumers [30,50,51,52]. For example, in the case of millennials, it has been noted that subjective norms are a factor in their motivation to consume wine in order to integrate into a group, spurred on by social pressure from relatives and friends and the condition of social acceptance [31].

The research work carried out in Italy by Scuderi et al. [26] shows how subjective norms are a crucial dimension in consumption behaviour as they measure how individuals identifying with people who tend to choose and consume Sicilian wine affects behavioural intention to consume this beverage. Therefore, it is supposed that if subjective norms relating to wine determine an individual’s consumption intention, they will also do so with local wine, and so the following hypothesis has been made:

**Hypothesis** **4** **(H4).***There is a positive relationship between subjective norms and local wine consumption intention*.

In a context related to the purchase of local foods, Shin and Hancer [14] confirmed how subjective norms are significant predictors of the formation of attitudes towards them, which in turn have an indirect influence on intention to buy. Furthermore, Scuderi et al. [26] studied intention to buy wine on the island of Sicily and confirmed that subjective norms correlate positively with perceived behavioural control. Therefore, in this study on local wine, it is proposed that there is an indirect relationship between subjective norms and local wine consumption intention through attitude towards them and perceived behavioural control.

**Hypothesis** **5** **(H5).***There is a positive relationship between subjective norms and attitude towards local wine consumption*.

**Hypothesis** **6** **(H6).***There is a positive relationship between subjective norms and perceived behavioural control*.

#### 2.1.4. Intention

Behavioural intention is defined as the likelihood of a person engaging in a particular behaviour [53]. The TPB postulates that intention reflects a person’s willingness to engage in objective behaviour. The results of research conducted by Capitello et al. [46] show that intention is a strong and significant deciding factor of behaviour as far as wine consumption is concerned. For their part, Maksan et al. [25] verified that intention has a positive impact on behaviour in terms of purchasing local wine. Therefore, the following is established:

**Hypothesis** **7** **(H7).***There is a positive relationship between local wine consumption intention and behaviour*.

### 2.2. Literature Review: Antecedents to Attitude towards Local Wine Consumption

In the context of consumer behaviour relating to local products, and despite the great explanatory power of the TPB, several researchers have suggested that more predictors should be added to increase its explanatory power [14]. The TPB is open to the inclusion of additional variables, provided that they contribute towards variation in the intention [40].

In this work, from the point of view of the Social Identity Theory, three antecedent factors to attitude towards behaviour have been added: personal norm, place satisfaction and place identity. The SIT [16] is defined as the interaction between personal and social identities. A person’s social identity is associated with their personal image and often each person joins different communities in order to acquire a strong self-identity. When the group’s interests come to the fore, individuals become depersonalised and adapt to the attitudes and behaviour of the prototypical members of their social group.

#### 2.2.1. Personal Norm

Personal norm is the feeling of moral obligation to carry out or abstain from specific actions [54]. Activated personal norms are experienced as feelings of moral obligations [55]. Thøgersen and Ölander [56] confirmed the influence of personal norms on the intention to buy organic food products. In a study of organic wine consumption [57], it was determined that there is strong evidence that personal norms can increase the likelihood of buying the product.

In this study, personal norms are examined as consumers’ moral responsibility to engage in proactive behaviour as far as consuming regional products is concerned and therefore the following hypothesis is put forward:

**Hypothesis** **8a** **(H8a).***There is a positive relationship between the personal norm and attitude towards local wine consumption*.

#### 2.2.2. Place Satisfaction

Place satisfaction refers to the judgement that individuals make of the perceived quality of an area/place, which satisfies or exceeds the needs and wishes of the local individuals [58].

Some studies have suggested the existence of a link between an individual’s satisfaction with a place and consumer behaviour that supports local products [59] and so the following hypothesis is proposed:

**Hypothesis** **8b** **(H8b).***There is a positive relationship between place satisfaction and attitude towards local wine consumption*.

#### 2.2.3. Place Identity

Place identity is a dimension of self-identity that defines ‘who we are’ in relation to ‘where we are’, that is, an individual’s sense of self in a physical environment [60,61,62,63,64,65]. This variable has also been studied from the perspective of national identity [66] referring to the extent to which people identify with and have a positive feeling of affiliation with their own country [16,67]. Zeugner-Roth, Žabkar and Diamantopoulos [19] confirmed that consumers with a strong national identity prefer products from their country of origin.

Therefore, it is intended to confirm the following hypothesis:

**Hypothesis** **8c** **(H8c).***There is a positive relationship between place identity and attitude towards local wine consumption*.

### 2.3. Influence of the Consumer’s Profile on Consumption Intention and Behaviour

On the other hand, many authors who have studied the consumption of local products have identified consumer profile personality variables such as ethnocentric or cosmopolitan orientation as influencers of consumer behaviour [19,68,69].

#### 2.3.1. Ethnocentrism

Consumer ethnocentrism derives from the concept of ethnocentrism in psychology which refers to people’s tendency to reject those who are different from them and to favour those that they perceive to be similar [70]. Consumer ethnocentrism is manifested by a social value that discriminates products from elsewhere [68,71,72] and can apply on a national, regional or local level [73]. Intention to buy local products is influenced both by ethnocentric tendencies as well as opinions of foreign products [74,75]. García, Chamorro and García [76] found that there is a positive relationship between consumer ethnocentrism and intention to buy regional wines. For their part, Fernández, Calvo, Bande, Artaraz and Galan [77] confirmed that consumer ethnocentrism influences actual purchases of food products that combine the characteristics of local, regional and traditional products. Therefore, the following hypotheses are proposed:

**Hypothesis** **9a** **(H9a).***There is a positive relationship between the ethnocentric profile and local wine consumption intention*.

**Hypothesis** **9b** **(H9b).***There is a positive relationship between the ethnocentric profile and local wine consumption*.

#### 2.3.2. Cosmopolitanism

A cosmopolitan orientation can be defined as an identity that emphasises receptiveness and the connection with the global community [78]. That is, the tendency to consider oneself a citizen of the world rather than a citizen of a specific country [79].

In this sense, the consumer’s cosmopolitanism reflects the extent to which they show an open mind towards foreign countries and cultures, appreciate the diversity brought about by the availability of products of different national and cultural origins, and have a positive disposition to consume products from different countries [80]. A cosmopolitan consumer will not necessarily consider foreign products to be intrinsically more attractive [81] but they may adopt a positive stance on the availability of products from cultures other than their own [80,82,83]. All of the above leads us to think that cosmopolitan consumers will neither have a positive impact on intention to consume local wine nor on its actual consumption and, therefore, the following hypotheses are proposed:

**Hypothesis** **10a** **(H10a).***There is a negative relationship between the cosmopolitan profile and local wine consumption intention*.

**Hypothesis** **10b** **(H10b).***There is a negative relationship between the cosmopolitan profile and local wine consumption*.

### 2.4. Theoretical Model of the Study

In accordance with the literary review carried out, Figure 1 shows an extended model of the TPB, integrating the Social Identity Theory to predict local wine consumption composed of 10 constructs. The 3 constructs on the left side of the figure (Personal Norm, Place Satisfaction and Place Identity) are adopted from the SIT while the 5 central constructs (Attitude towards local wine, Subjective Norms, Perceived Behavioural Control, Intention to local wine consumption and Local wine consumption) are derived from the original TPB model. Two consumer personality constructs have also been added in terms of Ethnocentrism and Cosmopolitan.

In this model it is proposed that local wine consumption depends on consumption intention and perceived behavioural control. On the other hand, it is suggested that attitude towards local wine consumption, subjective norms and perceived behavioural control have a direct and positive relationship with local wine consumption intention. Furthermore, the model includes the direct and positive relationships of subjective norms with perceived behavioural control and attitude towards consumption.

Finally, the model has been extended taking into account the Social Identity Theory by identifying the personal norm, place satisfaction and place identity as antecedents to attitude towards local wine consumption. It also takes into consideration the fact that ethnocentric and cosmopolitan profiles have an influence on local wine consumption intention and consumption.

### 2.5. Methodology

#### 2.5.1. Data Gathering and Measurements

To achieve the objectives set out, a self-administered online survey was conducted via a panel during the first half of August 2020. Eligibility criteria for participants in the study were to be over 18 years of age (legal age of consumption) and frequent consumers of wine (at least once every 15 days), selected from the general population of the Canary Islands by Toluna (www.toluna-group.com, accessed on 5 August 2020), an Internet panel company.

Due to the extensive availability on the Internet of different groups, Internet panels are increasingly being used as a viable means of obtaining data and are an efficient, low-cost solution [84]. Furthermore, Liu, Cella, Gershon, Shen, Morales, Riley and Hays [85] demonstrated that the representativeness of Internet data is comparable to probability sampling data from the general population.

To recruit the participants for the study, Toluna sent out emails inviting potential participants from their databases to sign up for the current study, following a selection process to ensure that they were eligible. The participants who completed the whole survey were compensated by means of incentives offered by the company, which has quality control procedures in place. The sample obtained was made up of 762 people, who had all completed the questionnaire and met the requirements.

The questionnaire was composed of 37 questions, divided into four sections; Filter and wine consumption habits, Intention and background (TPB), Consumer profile (cosmopolitan and ethnocentric) and Classification data. The items were measured by a seven point Likert scale about the level of agreement, where all items were anchored to a term. The items used for the measurement of the TPB-related constructs were those suggested by Maksan et al. [25]. The items used to measure the ethnocentric personality construct were adapted from the scale developed by Shimp and Sharma [86]. The measures of the cosmopolitan profile identification construct were adapted from the scale used by Cannon et al. [87].

#### 2.5.2. Data Processing

The profile of those surveyed can be seen in Table 1. Women make up 43.2% of the sample and men 56.8%. The participants were from the following age groups: 9.4% were 18 to 24 years of age; 22.8% were in the 25 to 34 age group; 26.4% were 35 to 44 years of age; 23.4% were aged from 45 to 54; and 18.0% were 55 or over. In terms of their level of education, 0.7% of the sample stated that their level of education was basic or primary, 36.5% had completed secondary education, baccalaureate or vocational training, and 62.9% had a university degree. Participants were resident on the following islands: 43.3% of responses were obtained from the island of Tenerife; 7.5% from La Palma; 3.0% from La Gomera; 3.7% from El Hierro; 28.3% from Gran Canaria; 8.1% from Lanzarote; and 6.0% from Fuerteventura. Social status was measured based on the respondents’ own perception of their income and the following results were obtained: 17.7% of the sample considered that their income was below average; 61.3% considered their income to be average; and 19.7% considered their income to be above average. The resulting structure of the sample shows that 22.8% consume wine once every 15 days, 31.7% at least once a week, 36.9% several times a week and 8.6% admit to drinking wine on a daily basis.

To verify that the sample size was sufficient for the analyses to be carried out, G*Power [88] was used which suggests that a minimum sample of 172 individuals is required to test the proposed model (10 constructs) and achieve a statistical power of 0.95. Therefore, it can be confidently conclude that the size of the sample used (762) is much greater than that required for the purposes of this study.

To analyse the proposed theoretical model and test the hypotheses, the partial least squares (PLS-SEM) technique was applied using the software Smart PLS v.3.3.3 [89]. For the calculations, a weighting vector was used with the aim of adjusting the sample to the population structure according to the size of the population of each island.

## 3. Results

### 3.1. Descriptive Analysis

Table 2 shows the results of the descriptive analysis (mean and standard deviation) of the construct items of the proposed model. It can be seen that there is an average local wine consumption intention among the population researched, the mean of the construct items being between 4.74 and 4.96 on a scale from 1 (totally disagree) to 7 (totally agree). Consumer behaviour is below the mean of the scale, ranging from 3.78 to 3.90.

The items of the antecedent and predictive latent variables relating to consumption intention show values slightly above average. The mean values of the indicators relating to personal attitude towards local wine range from 4.73 to 5.06. Subjective norms items are between 4.19 and 4.86. Wine consumers also exhibit an average perceived behavioural control with indicator means ranging from 4.33 to 5.13.

The latent variables proposed in the model as antecedents to attitude towards local wine obtained values a little higher than the mean. The personal norm or moral commitment to defend what is local obtained average values in its indicators, which are also slightly higher than the mean at 4.43 to 5.14. However, both place satisfaction and place identity obtained slightly higher values. The items ‘Satisfaction with the tranquillity of the Canary Islands’ and ‘Satisfaction with the living environment on the islands’ obtained means of between 5.07 and 5.68. Regarding the place identity indicators, they have means ranging from 5.25 to 5.42.

There is a level of self-perception of the ethnocentric consumer which shows a certain extent of support for local products with means of between 4.20 and 5.03. The perception of open-mindedness and willingness to consume products from elsewhere obtained results a little higher than those of the ethnocentric consumer, with means ranging from 5.11 to 5.45.

### 3.2. Evaluation of the Overall Model

The results revealed SRMR model fit values of 0.058, and a value of less than 0.08 can be considered acceptable for PLS-SEM [90]. It has also been confirmed that there are no indications of multicollinearity among the antecedent variables of each of the endogenous constructs as all of the VIF (variance inflation factor) values are less than 5. Finally, common method bias (CMB) does not appear to be an issue as there was no single factor (five different factors represented 69.4% of the total variance) and the first factor did not represent the majority of the variance (accounting for 43.9%) [91].

### 3.3. Evaluation of the Measurement Model

Table 2 shows the following: all of the loadings of the items in the final measurement model are greater than 0.707 [92]; all of the Cronbach’s alpha and composite reliability values [93] are above the minimum cut-off point of 0.70 [94]; and the Average Variance Extracted of each latent variable is greater than 0.5 [94]. Likewise, the results in Table 3 show that the constructs examined exceeded the heterotrait–monotrait (HTMT) requirements of the correlations (values less than 0.85) [95]. Therefore, the measurement model is considered to be satisfactory and provided sufficient proof in terms of reliability and convergent and discriminant validity.

### 3.4. Evaluation of the Structural Model

The path coefficients show the estimates of the relationships in the structural model. The significance of the effects was evaluated using bootstrapping [96]: (1) One-tailed Student’s t distribution test with *n* − 1 degrees of freedom (*n* = 5000 subsamples); (2) Confidence interval analysis [97].

As can be seen in Table 4 and Figure 2, perceived behavioural control has the greatest direct significant effect on local wine consumption behaviour (H3: β = 0.457, *p* < 0.001, f^2^ = 0.412), with a high f^2^ indicator (the f^2^ effect size evaluates the degree to which an exogenous construct contributes to explaining a specific endogenous construct in terms of R^2^ [98]).

Consumption intention is also significant and closely related to local wine consumption behaviour (H7: β = 0.440, *p* < 0.001, f^2^ = 0.361). Furthermore, although the ethnocentric consumer profile has a significant positive direct relationship with behaviour, the path coefficient is very small, as is the f^2^ effect size (H9b: β = 0.099, *p* < 0.001, f^2^ = 0.023). The cosmopolitan profile has a significant direct negative relationship with consumption behaviour, although the f^2^ effect size is small (H10b: β = −0.093, *p* < 0.001, f^2^ = 0.026).

Hypotheses H7, H3, H9b and H10b are also supported empirically, although H9b and H10b are the weakest due to their low contribution and small effect size, meaning that their influence on local wine consumption behaviour is limited.

Regarding the hypotheses put forward concerning the direct relationship of the antecedent latent variables of local wine consumption intention, it can be seen that attitude towards consumption has the greatest significant effect with a small effect size (H1: β = 0.351, *p* < 0.001, f^2^ = 0.064). The ethnocentric consumer profile also has a significant positive relationship with local wine consumption intention (H9a: β = 0.160, *p* < 0.001, f^2^ = 0.036). Likewise, perceived behavioural control has a significant positive relationship with consumption intention (H2: β = 0.151, *p* < 0.001, f^2^ = 0.015).

In turn, subjective norms have a positive direct relationship with consumption intention but with a lower significance and a very small effect size (H4: β = 0.114, *p* < 0.05, f^2^ = 0.006). The cosmopolitan consumer profile has a positive relationship with local wine consumption intention, contrary to what was expected, although the significance is low and the effect size small (H10a: β = 0.072, *p* < 0.05, f^2^ = 0.009).

There is sufficient evidence to accept research hypotheses H1, H2, H4 and H9a, although H4′s contribution is very small, of low significance and has a small effect size, meaning that its influence on local wine consumption intention is limited.

However, there is no evidence to support H10a since, although the hypothesis postulated a negative relationship, a positive relationship is observed, albeit quite weak.

Attitude towards local wine consumption is directly and positively influenced by subjective norms (H5: β = 0.705, *p* < 0.001, f^2^ = 1.143), the personal norm (H8a: β = 0.104, *p* < 0.001, f^2^ = 0.025), and place identity (H8c: β = 0.099, *p* < 0.01, f^2^ = 0.017). Therefore, hypotheses H5, H8a and H8c are supported empirically, although the latter has a small effect size. On the contrary, in the case of hypothesis H8c, which postulated a direct relationship between place satisfaction and attitude towards local wine, there is no evidence to accept it.

Finally, evidence can be found to support hypothesis H6, which postulates a positive relationship between subjective norms and perceived behavioural control (H6: β = 0.797, *p* < 0.001, f^2^ = 1.774), with a strong influence and a large effect size.

The coefficient of determination (R^2^) represents a predictive power measurement that indicates the amount of variance of a construct which is explained by the predictive values of said endogenous construct in the model. The proposed model explains 69.5% of local wine consumption behaviour, 49.4% of consumption intention, 72.2% of attitude towards local wine, and 63.5% of perceived behavioural control (Table 4).

Furthermore, the Stone–Geisser test was used as a criterion to measure the predictive relevance of the constructs [99,100] and in Table 4 it can be seen that the Q^2^ values are greater than zero, which indicates that the model has predictive potential.

### 3.5. Importance–Performance Map Analysis

To expand on the results of the PLS-SEM, an Importance–Performance Map Analysis (IPMA) was carried out [101] which consists of showing on the horizontal axis the value of the total effects of the constructs in the structural model obtained from the path coefficients (importance) and, on the vertical axis the performance of each construct obtained by rescaling the means of the latent variable scores to a range from 0 to 100.

Table 5 shows the IPMA results and can be seen that the effect of the variable SN on behaviour (0.576) and intention (0.482), and of the variable PBC on behaviour (0.523), behavioural intention (0.440) and attitude towards intention (0.351), are those that have the greatest effect when explaining the constructs.

However, if the performance is observed, it can be confirmed that the variables SN (59.95%) and PBC (59.71%) have more margin for improvement. Therefore, efforts to improve local wine consumption intention and behaviour must focus on SN and PBC.

## 4. Discussion

The results of this research confirm the great influence of behavioural intention on local wine consumption as postulated by the TPB [40]. These results corroborate the findings of Capitello et al. [46] and Maksan et al. [25] who showed that intention is a strong and significant deciding factor of behaviour as far as wine consumption is concerned.

This study also confirms the findings of other studies such as those of Ho Shin and Hancer [14] who confirmed how perceived behavioural control as far as local food products are concerned has an influence on intention to buy them.

More specifically, and relating to wine consumption and local identity, the results of this study coincide with those of Capitello, Agnoli and Begalli [46] who proved the existence of a significant correlation between perceived behavioural control and wine consumption intention.

Subjective norms has also turned out to be a factor that has an influence on local wine consumption intention, coinciding with the findings of the majority of studies that looked into this dimension, such as the work of Capitello et al. [30]; St James and Christodoulidou [50], Thompson and Vourvachis [51], van Zanten [52] and Scuderi et al. [26].

This research also confirms the influence of the ethnocentric consumer profile on the selection of and preference for local wine, coinciding with the findings of García et al. [76] who revealed a positive relationship between consumer ethnocentrism and intention to buy regional wines.

Furthermore, the results of the study confirm the suggestions of Caldwell, Blackwell and Tulloch [82] and Skrbis et al. [83] in the sense that a cosmopolitan consumer may also conscientiously decide to consume products from cultures other than their own, revealing a negative relationship between the cosmopolitan profile and intention to consume local wine.

### 4.1. Contributions to the Literature

This research has made several relevant contributions to the literature. Firstly, constructs from the Social Identity Theory have been included in the TPB which have contributed to explaining local wine consumption behaviour. Secondly, the influencing constructs of the cosmopolitan and ethnocentric consumer personalities have been added to the theoretical model. Finally, the high R^2^ of the model supported the constructs used to explain local wine consumption intention. The additional constructs that have been included could be useful in other attempts to explain intention to consume local products.

### 4.2. Management Implications

These findings may be used as a reference to develop impulse strategies for local wine and agricultural product consumption. First and foremost, it must be pointed out that, in view of the importance of perceived social pressure to engage in behaviour relating to the consumption of local agricultural products such as wine, public campaigns should be reinforced in order to emphasise the importance of individual behaviour as far as local products are concerned. It would be necessary to highlight that the participation of opinion leaders and social influencers in such campaigns is fundamental due to their relevant role in terms of subjective norms. Likewise, perceived behavioural control is one of the variables that should be acted upon due to its potential to influence and improve. In this sense, extensive information on the characteristics of the product and its availability in the sales channels are key elements.

## 5. Conclusions

In this research, an extended model of the TPB has been developed to predict behaviour in the area of choosing local wine. Different studies have confirmed the importance of consuming local products as far as protecting the local economy and jobs are concerned, as well as in terms of preserving the rural landscape and environment and reducing environmental impacts.

According to the results obtained, local wine consumption is mainly explained by intention and perceived behavioural control, although the ethnocentric personality also has a positive influence and the cosmopolitan personality a negative one. On the other hand, intention is affected above all by attitude towards local products, perceived behavioural control and the ethnocentric personality.

Attitude towards local products, for its part, is mainly influenced by subjective norms, although it is also affected by place identity and the personal norm.

This study has therefore developed and corroborated an explanatory model which, taking the TPB as a starting point, has increased its explanatory power, adding key factors from the Social Identity Theory and characteristics of the ethnocentric and cosmopolitan personalities, and improving the theoretical framework used to explain consumption behaviour relating to local products such as wine.

### Limitations

Firstly, this study focused on local wine consumption behaviour in a specific region (Canary Islands), meaning that new research can be justified that looks into intentions and behaviour related to the selection of local wine in other places and regions, with the aim of determining whether the patterns and results are repeated or depend on factors such as culture or other influences.

Secondly, additional variables that expand on the TPB have been included in this study. However, other variables may be used that could improve the predictive value of the model. Along the same lines, one limitation of this study is the fact that the respondents belonged to a panel and bias may be an issue.

## Figures and Tables

**Figure 1 foods-10-02187-f001:**
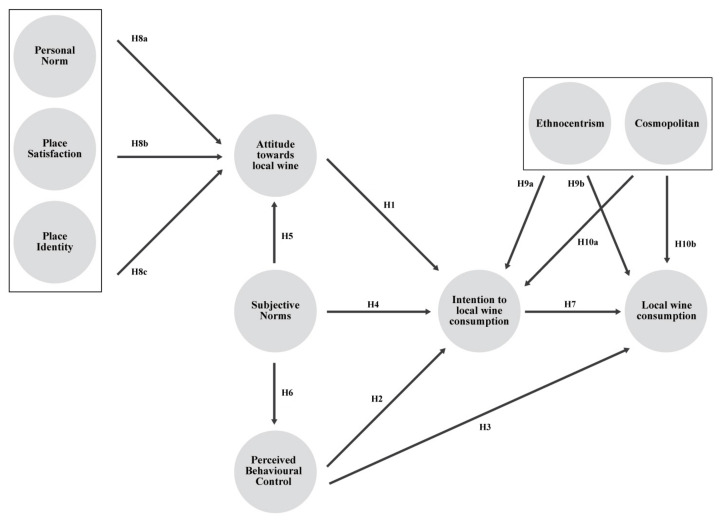
Proposed Theoretical Model.

**Figure 2 foods-10-02187-f002:**
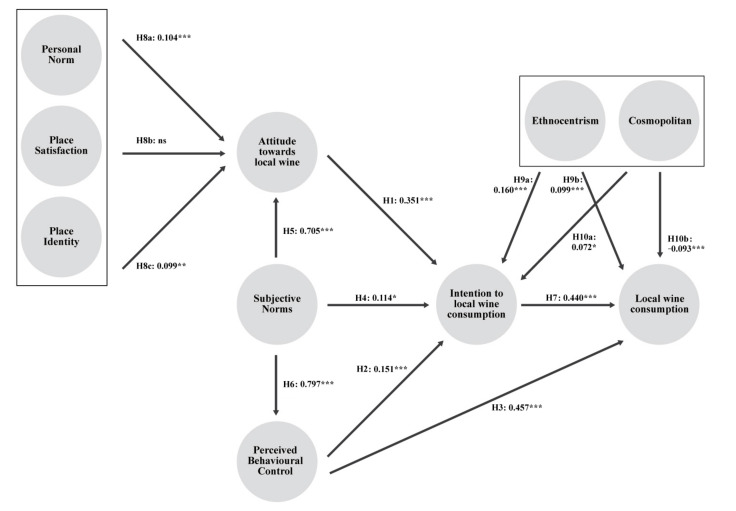
Results of the analysis. Significance level: * *p* < 0.5; ** *p* < 0.1; ***; *p* < 0.01; ns: non-significant.

**Table 1 foods-10-02187-t001:** Profile of the respondents.

Sex	
Female	43.2%
Male	56.8%
**Age**	
18–24 years	9.4%
25–34 years	22.8%
35–44 years	26.4%
45–54 years	23.4%
≥55 years	18.0%
**Level of education**	
Basic/primary	0.7%
Secondary/baccalaureate/vocational training	36.5%
University	62.9%
**Island of residence**	
Tenerife	43.3%
La Palma	7.5%
La Gomera	3.0%
El Hierro	3.7%
Gran Canaria	28.3%
Lanzarote	8.1%
Fuerteventura	6.0%
**Income**	
Below average	17.7%
Average	61.3%
Above average	19.7%
Not known	1.3%
**Frequency of wine consumption**	
Once every 15 days	22.8%
Once a week	31.7%
Several times a week	36.9%
Daily	8.6%
**Total**	762

**Table 2 foods-10-02187-t002:** Descriptive analysis and results of the evaluation of the measurement model.

Constructs/Associated Items		Mean	Standard Deviation	Loading	Cronbach’s Alpha	Composite Reliability	Average Variance Extracted
**B**	**Behaviour**				0.959	0.974	0.925
B1	I usually order or buy Canarian wine.	3.89	1.946	0.968			
B2	I regularly have Canarian wine on my table.	3.78	1.943	0.957			
B3	I usually choose Canarian wine from a selection of different wines.	3.90	1.908	0.960			
**IN**	**Intention**				0.931	0.956	0.879
IN1	I intend to buy or consume Canarian wine on a regular basis.	4.74	1.532	0.947			
IN2	I am determined to buy or consume Canarian wine.	4.96	1.520	0.924			
IN3	I will probably buy or consume Canarian wine regularly.	4.81	1.505	0.941			
**AT**	**Attitude**				0.907	0.942	0.843
AT1	I find buying Canarian wine on a regular basis satisfying.	4.73	1.512	0.924			
AT2	Buying Canarian wine regularly makes me feel positive.	4.74	1.455	0.917			
AT3	I feel proud to buy Canarian wine.	5.06	1.406	0.914			
**SN**	**Subjective Norms**				0.838	0.902	0.755
SN1	My friends and relatives approve of me consuming Canarian wine.	4.86	1.448	0.868			
SN2	I make a good impression on my friends and colleagues by ordering Canarian wine.	4.68	1.422	0.880			
SN3	My friends and acquaintances regularly buy Canarian wine.	4.19	1.671	0.859			
**PBC**	**Perceived Behavioural Control**				0.798	0.881	0.714
PBC1	Whether or not I regularly buy Canarian wine is my own decision.	5.13	1.502	0.723			
PBC2	My friends and acquaintances regularly buy Canarian wine.	4.33	1.720	0.901			
PBC3	I find it easy to buy Canarian wine.	4.41	1.783	0.899			
**PN**	**Personal Norm**				0.830	0.898	0.746
PN1	I have a responsibility to consume Canarian products, even if it means I have to sacrifice personal benefits.	4.43	1.541	0.834			
PN2	I feel morally obliged to defend Canarian products.	4.91	1.523	0.897			
PN3	Everyone has a responsibility to favour goods produced on the islands.	5.14	1.331	0.859			
**PS**	**Place Satisfaction**				0.811	0.888	0.728
PS1	Satisfaction with the tranquillity of the Canary Islands.	5.68	1.288	0.895			
PS2	Satisfaction with the level of development of the islands.	5.07	1.450	0.730			
PS3	Satisfaction with the living environment on the islands.	5.65	1.263	0.922			
**PI**	**Place Identity**				0.900	0.937	0.833
PI1	I feel that I belong here and it is part of my identity.	5.26	1.533	0.928			
PI2	I like living here and I feel connected to the place.	5.42	1.362	0.936			
PI3	It is the best place to do the things that I enjoy.	5.25	1.381	0.872			
**ETH**	**Ethnocentric consumer**				0.755	0.854	0.662
ETH1	Canarians should always buy products made in the Canary Islands rather than imported goods.	4.52	1.523	0.851			
ETH2	There should be very little business and purchasing of products from other countries unless it is necessary.	4.20	1.517	0.708			
ETH3	It may cost me more in the long term, but I prefer to back Canarian products.	5.03	1.251	0.872			
**COS**	**Cosmopolitan consumer**				0.853	0.909	0.769
COS1	I am open-minded towards foreign countries and cultures.	5.45	1.327	0.872			
COS2	I appreciate the diversity of products of different origins.	5.41	1.235	0.907			
COS3	I have a positive attitude towards the consumption of products from different countries.	5.11	1.328	0.851			

Scale: from 1 (Totally disagree) to 7 (Totally agree).

**Table 3 foods-10-02187-t003:** Discriminant validity.

Constructs	ATT	B	ETH	COS	IN	PBC	PI	PN	PS	SN
Heterotrait–Monotrait Ratio (HTMT)										
ATT										
B	0.790									
ETH	0.604	0.561								
COS	0.351	0.177	0.226							
IN	0.727	0.775	0.538	0.308						
PBC	0.876	0.824	0.539	0.323	0.684					
PI	0.623	0.542	0.453	0.263	0.515	0.657				
PN	0.613	0.483	0.695	0.156	0.487	0.527	0.621			
PS	0.543	0.393	0.351	0.329	0.427	0.536	0.726	0.587		
SN	0.954	0.791	0.597	0.338	0.709	0.962	0.619	0.598	0.530	

**Table 4 foods-10-02187-t004:** Results of the hypothesis test, Variance Breakdown, Predictive Relevance redundancy and Effect size. * *p* < 0.5; ** *p* < 0.1; ***, *p* < 0.01; ns: non-significant.

		**Path Coefficient**	**Sig.**	**T Statistics**	**Confidence Intervals**	**Confidence Interval Bias**	**Supported**	**Variable Correlation**	**Coefficient of Determination (R^2^)**	**Predictive Relevance (Q^2^)**	**Effect Size (f^2^)**
	**Behaviour**								0.695	0.636	
H7	IN -> B	0.440	***	11.777	[0.378; 0.501]	[0.377; 0.5]	Yes/Yes	0.734	0.323		0.361
H3	PBC -> B	0.457	***	15.028	[0.406; 0.507]	[0.405; 0.507]	Yes/Yes	0.739	0.338		0.412
H9b	ETH -> B	0.099	***	3.490	[0.052; 0.145]	[0.052; 0.145]	Yes/Yes	0.500	0.050		0.023
H10b	COS -> B	−0.093	***	3.587	[−0.136; -0.05]	[−0.137; -0.052]	Yes/Yes	0.167	−0.016		0.026
	**Intention**								0.494	0.428	
H1	ATT -> IN	0.351	***	6.898	[0.268; 0.435]	[0.268; 0.435]	Yes/Yes	0.668	0.234		0.064
H4	SN -> IN	0.114	*	2.048	[0.019; 0.203]	[0.022; 0.206]	Yes/Yes	0.627	0.071		0.006
H2	PBC -> IN	0.151	***	3.196	[0.077; 0.229]	[0.074; 0.227]	Yes/Yes	0.596	0.090		0.015
H9a	ETH -> IN	0.160	***	3.974	[0.095; 0.226]	[0.094; 0.225]	Yes/Yes	0.485	0.078		0.036
H10a	COS -> IN	0.072	*	2.076	[0.015; 0.129]	[0.012; 0.127]	Yes/Yes	0.283	0.020		0.009
	**Attitude**								0.722	0.604	
H5	SN -> ATT	0.705	***	23.759	[0.653; 0.751]	[0.654; 0.751]	Yes/Yes	0.832	0.587		1.143
H8c	PI -> ATT	0.099	**	2.652	[0.04; 0.162]	[0.039; 0.162]	Yes/Yes	0.567	0.056		0.017
H8a	PN -> ATT	0.104	***	3.902	[0.061; 0.148]	[0.06; 0.147]	Yes/Yes	0.533	0.055		0.025
H8b	PS -> ATT	0.051	ns	1.494	[−0.004; 0.106]	[−0.003; 0.108]	No/No	0.472	0.024		0.005
	**Perceived Behavioural Control**								0.635	0.447	
H6	SN -> PBC	0.797	***	50.265	[0.77; 0.822]	[0.767; 0.82]	Yes/Yes	0.797	0.635		1.744

**Table 5 foods-10-02187-t005:** Importance–Performance Map Analysis results.

	Behaviour	Intention	Performance
ATT	0.15	0.35	64.12
ETH	0.17	0.16	61.70
COS	−0.06	0.07	72.52
IN	0.44		63.89
PBC	0.52	0.15	59.71
PI	0.02	0.04	71.94
PN	0.02	0.04	64.27
PS	0.01	0.02	75.45
SN	0.58	0.48	59.95

## Data Availability

The data presented in this study are available on request from the corresponding author.

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
