# Peer review of "An Extended Model of the Theory of Planned Behaviour to Predict Local Wine Consumption Intention and Behaviour"

_foods, 2021, doi:10.3390/foods10092187_

Round 1

Reviewer 1 Report

This paper aims to study the background of local wine consumption behaviour, using an extended model of the Theory of Planned Behaviour, by means of a survey conducted in the Canary Islands with a sample of 762 people.

Results show that local wine consumption is mainly explained by intention and perceived behavioural control, although the ethnocentric personality also has a positive influence. Moreover, the personal norm and place identity are also confirmed to be  related to attitude towards such behavior.

It is an interesting issue for Foods.

However, author should improve manuscript as follows:

  1. In the Introduction section, authors have to better identify the literature gap that is to be studied.  What is the novelty of the study?  What value does it contribute to the existing literature?  The justifications of the study are suggested to clearly present in the introduction.
  2. Since authors adopt the TPB, they should improve the 2.1 paragraph, inasmuch it is too synthetic or at least it seems detached from the following sub-paragraphs.
  3. Also in Materials and Methods section, authors should reorder or renumber hypotheses or sub-paragraphs: the first hypothesis which appears in the text is the H2, followed by H3, H6, H7, etc.
  4. In Methodology section, Authors should include some information about the questionnaire adopted for the survey (i.e. number of sections, what information are required by each section, etc.).
  5. Why don’t Authors adopt items of CEESCALE or CETSCALE in order to measure consumer ethnocentrism?
  6.  The authors should better explain the 10-constructs model in 2.4 paragraph for the readers. The 10 adopted constructs and their items are clear only in Tab. 2.   

I don't feel qualified to judge about the English Language and Style.

So, I recommend the publication with major revisions.

Reviewer 2 Report

The authors present a very well written and comprehensive study about the behavior science aspect of wine consumption. The background information is well laid out and the study is has an appropriate design. There a only a few minor issues that need to be addressed.

Specific comments

Line 80   I see that the hypotheses are not in order but this is a little confusing. Is there any way to renumber them or change the order? It is fairly confusing right now.

Line 241 The figure is a little blurry and the font is too small. Please revise.

Line 259 Are those percentages representative of the wine consumer market of the Canary Islands?

Line 310 Please reformat the table so the headings can be read.

Line 337 Please reformat the table so the headings can be read. Also, please do not use abbreviations or symbols without explanation.

Line 338 Consider rephrasing to "as can be seen in Table 4".

Line 352 Please check throughout the manuscript and avoid we-statements.

Line 392 This applies to almost all tables and figures; please do not use abbreviations without explanation. Every table or figure should be able to stand alone and still be understandable.

Line 434 I am not sure that this discussion paragraph is fully supported by the data. Management implications are not necessarily an appropriate part of a scientific evaluation.

Reviewer 3 Report

The manuscript by Sabina del Castillo and coworkers on an extended model of the TPB to predict local wine consumption reports potential consumer-inherent factors tha may explain intention to consume and consumption of locally produced wine. The work is of interest to the readers in my opinion and has clearly laid out research hypotheses.

In general, the manuscript reads well and covers the relevant background literature. I only have some minor comments that would help improve the manuscript even more:

1) when listing the hypotheses, it might be worthwhile numbering them in the order they are presented/stated? Right now, H2 through H7 are described before H1.

2) The size and quality of figure 1 needs to be improved - the font size is too small to be readable and also very pixelated. I would also suggest to add the degree of correlation (positive or negative) to each arrow?

3) More details about the actual questionnaire are needed, e.g., how was a "frequent consumer of wine" defined? when was the study conducted? How long was it fielded? How were questions answered - Likert scale? If so, how many options, e.g., 5-point, 7-point? Were all points anchored with a term or just the endpoints? How many people were recruited, excluded, etc? What inclusion and exclusion criteria were used? What were the eligibility criteria? Why were the different Canarian islands analzyed separately? Some justification/explanation for this is needed.

4) For the age split - it seems that an analysis by generational cohort would be better, especially as numerous wine consumer research demonstrated generational differences in wine consumption between e.g., Genz Z, Millennials, Gen X and Boomers; maybe even split Millennials into younger and older ones? Also, how does the age distribution of the recruited participants reflect wine consumers in the Canary Islands?

5) for the wine frequency consumption question - the options seem to lack an option for between every 7 and 14 days and less than every 15 days?

6) Assuming a 7-point Likert scale was used, the sentence in L290-291 is IMO not correct - a value of 3.8-3.9 on a 7-point scale would be just above the average, so the authors cannot state local wine consumption is relatively uncommon?

7) More details about the statistical analysis need to be included - e.g., statistical software used, etc.

Round 2

Reviewer 1 Report

Dear authors, I appreciated your efforts to improve manuscript.